# Multi-Agent Common Knowledge Reinforcement Learning

**Christian A. Schroeder de Witt**[*][†]    **Jakob N. Foerster**[*][†]

**Gregory Farquhar**[†]    **Philip H. S. Torr**[†]    **Wendelin Böhmer**[†]    **Shimon Whiteson**[†]

## Abstract

Cooperative multi-agent reinforcement learning often requires decentralised policies, which severely limit the agents' ability to coordinate their behaviour. In this paper, we show that common knowledge between agents allows for complex decentralised coordination. Common knowledge arises naturally in a large number of decentralised cooperative multi-agent tasks, for example, when agents can reconstruct parts of each others' observations. Since agents can independently agree on their common knowledge, they can execute complex coordinated policies that condition on this knowledge in a fully decentralised fashion. We propose *multi-agent common knowledge reinforcement learning* (MACKRL), a novel stochastic actor-critic algorithm that learns a hierarchical policy tree. Higher levels in the hierarchy coordinate groups of agents by conditioning on their common knowledge, or delegate to lower levels with smaller subgroups but potentially richer common knowledge. The entire policy tree can be executed in a fully decentralised fashion. As the lowest policy tree level consists of independent policies for each agent, MACKRL reduces to independently learnt decentralised policies as a special case. We demonstrate that our method can exploit common knowledge for superior performance on complex decentralised coordination tasks, including a stochastic matrix game and challenging problems in StarCraft II unit micromanagement.

## 1   Introduction

Cooperative multi-agent problems are ubiquitous, for example, in the coordination of autonomous cars (Cao et al., 2013) or unmanned aerial vehicles (Pham et al., 2018; Xu et al., 2018). However, how to learn control policies for such systems remains a major open question.

*Joint action learning* (JAL, Claus & Boutilier, 1998) learns centralised policies that select joint actions conditioned on the global state or joint observation. In order to execute such policies, the agents need access to either the global state or an instantaneous communication channel with sufficient bandwidth to enable them to aggregate their individual observations. These requirements often do not hold in practice, but even when they do, learning a centralised policy can be infeasible as the size of the joint action space grows exponentially in the number of agents. By contrast, *independent learning* (IL, Tan, 1993) learns fully decentralisable policies but introduces nonstationarity as each agent treats the other agents as part of its environment.

These difficulties motivate an alternative approach: centralised training of decentralised policies. During learning the agents can share observations, parameters, gradients, etc. without restriction but the result of learning is a set of decentralised policies such that each agent can select actions based only on its individual observations.

---

[*]Equal contribution. Correspondence to Christian Schroeder de Witt <cs@robots.ox.ac.uk>
[†]University of Oxford, UK

While significant progress has been made in this direction (Rashid et al., 2018; Foerster et al., 2016, 2017, 2018; Kraemer & Banerjee, 2016; Jorge et al., 2016), the requirement that policies must be fully decentralised severely limits the agents' ability to coordinate their behaviour. Often agents are forced to ignore information in their individual observations that would in principle be useful for maximising reward, because acting on it would make their behaviour less predictable to their teammates. This limitation is particularly salient in IL, which cannot solve many coordination tasks (Claus & Boutilier, 1998).

*Common knowledge* for a group of agents consists of facts that all agents know and "each individual knows that all other individuals know it, each individual knows that all other individuals know that all the individuals know it, and so on" (Osborne & Rubinstein, 1994). This may arise in a wide range of multi-agent problems, e.g., whenever a reliable communication channel is present. But common knowledge can also arise without communication, if agents can infer some part of each other's observations. For example, if each agent can reliably observe objects within its field of view and the agents know each other's fields of view, then they share common knowledge whenever they see each other. This setting is illustrated in Figure 1 and applies to a range of real-world scenarios, for example, to robo-soccer (Genter et al., 2017), fleets of self-driving cars and multi-agent StarCraft micromanagement (Synnaeve et al., 2016).

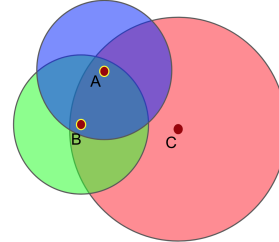

Figure 1: Three agents and their fields of view. A and B's locations are common knowledge to A and B as they are within each other's fields of view. Although C can see A and B, it shares no common knowledge with them.

In the absence of common knowledge, complex decentralised coordination has to rely on implicit communication, i.e., observing each other's actions or their effects (Heider & Simmel, 1944; Rasouli et al., 2017). However, implicit communication protocols for complex coordination problems are difficult to learn and, as they typically require multiple timesteps to execute, can limit the agility of control during execution (Tian et al., 2018). By contrast, coordination based on common knowledge is simultaneous, that is, does not require learning communication protocols (Halpern & Moses, 2000).

In this paper, we introduce *multi-agent common knowledge reinforcement learning* (MACKRL), a novel stochastic policy actor-critic algorithm that can learn complex coordination policies end-to-end by exploiting common knowledge between groups of agents at the appropriate level. MACKRL uses a hierarchical policy tree in order to dynamically select the right level of coordination. By conditioning joint policies on common knowledge of groups of agents, MACKRL occupies a unique middle ground between IL and JAL, while remaining fully decentralised.

Using a proof-of-concept matrix game, we show that MACKRL outperforms both IL and JAL. Furthermore, we use a noisy variant of the same matrix game to show that MACKRL can exploit a weaker form of group knowledge called *probabilistic common knowledge* (Krasucki et al., 1991), that is induced by agent beliefs over common knowledge, derived from noisy observations. We show that MACKRL's performance degrades gracefully with increasing noise levels.

We then apply MACKRL to challenging StarCraft II unit micromanagement tasks (Vinyals et al., 2017) from the *StarCraft Multi-Agent Challenge* (SMAC, Samvelyan et al., 2019). We show that simultaneous coordination based on pairwise common knowledge enables MACKRL to outperform the state-of-the-art algorithms COMA (Foerster et al., 2018) and QMIX (Rashid et al., 2018) and provide a variant of MACKRL that scales to tasks with many agents.

## 2   Problem Setting

Cooperative multi-agent tasks with $n$ agents $a \in \mathcal{A}$ can be modelled as *decentralised partially observable Markov decision processes* (Dec-POMDPs, Oliehoek et al., 2008). The state of the system is $s \in \mathcal{S}$. At each time-step, each agent $a$ receives an observation $z^a \in \mathcal{Z}$ and can select an action $u_{\text{env}}^a \in \mathcal{U}_{\text{env}}^a$. We use the env-subscript to denote actions executed by the agents in the environment, as opposed to latent 'actions' that may be taken by higher-level controllers of the hierarchical method introduced in Section 3. Given a joint action $\mathbf{u}_{\text{env}} := (u_{\text{env}}^1, \ldots, u_{\text{env}}^n) \in \mathcal{U}_{\text{env}}$, the discrete-time system dynamics draw the successive state $s' \in \mathcal{S}$ from the conditional distribution $P(s'|s, \mathbf{u}_{\text{env}})$ and yield a cooperative reward according to the function $r(s, \mathbf{u}_{\text{env}})$.

The agents aim to maximise the discounted return $R_t = \sum_{l=0}^{H} \gamma^l\, r(s_{t+l}, \mathbf{u}_{t+l,\text{env}})$ from episodes of length $H$. The joint policy $\pi(\mathbf{u}_{\text{env}}|s)$ is restricted to a set of decentralised policies $\pi^a(u^a_{\text{env}}|\tau^a_t)$ that can be executed independently, i.e., each agent's policy conditions only on its own action-observation history $\tau^a_t := [z^a_0, u^a_0, z^a_1, \dots, z^a_t]$. Following previous work (Rashid et al., 2018; Foerster et al., 2016, 2017, 2018; Kraemer & Banerjee, 2016; Jorge et al., 2016), we allow decentralised policies to be learnt in a centralised fashion.

**Common knowledge** of a group of agents $\mathcal{G}$ refers to facts that all members know, and that "each individual knows that all other individuals know it, each individual knows that all other individuals know that all the individuals know it, and so on" Osborne & Rubinstein (1994). Any data $\xi$ that are known to all agents before execution/training, like a shared random seed, are obviously common knowledge. Crucially, every agent $a \in \mathcal{G}$ can deduce the same *history of common knowledge* $\tau^{\mathcal{G}}_t$ from its own history $\tau^a_t$ and the commonly known data $\xi$, that is, $\tau^{\mathcal{G}}_t := \mathcal{I}^{\mathcal{G}}(\tau^a_t, \xi) = \mathcal{I}^{\mathcal{G}}(\tau^{\bar{a}}_t, \xi), \forall a, \bar{a} \in \mathcal{G}$. Furthermore, any actions taken by a policy $\pi^{\mathcal{G}}(\mathbf{u}^{\mathcal{G}}_{\text{env}}|\tau^{\mathcal{G}}_t)$ over the group's joint action space $\mathcal{U}^{\mathcal{G}}_{\text{env}}$ are themselves common knowledge, if the policy is deterministic or pseudo-random with a shared random seed and conditions only on the common history $\tau^{\mathcal{G}}_t$, i.e. the set formed by restricting each transition tuple within the joint history of agents in $\mathcal{G}$ to what is commonly known in $\mathcal{G}$ at time $t$. Common knowledge of subgroups $\mathcal{G}' \subset \mathcal{G}$ cannot decrease, that is, $\mathcal{I}^{\mathcal{G}'}(\tau^a_t, \xi) \supseteq \mathcal{I}^{\mathcal{G}}(\tau^a_t, \xi)$.

Given a Dec-POMDP with noisy observations, agents in a group $\mathcal{G}$ might not be able to establish true common knowledge even if sensor noise properties are commonly known (Halpern & Moses, 2000). Instead, each agent $a$ can only deduce its own *beliefs* $\tilde{\mathcal{I}}^{\mathcal{G}}_a(\tilde{\tau}^a_t)$ over what is commonly known within $\mathcal{G}$, where $\tilde{\tau}^a_t$ is the agent's belief over what constitutes the groups' common history. Each agent $a$ can then evaluate its own *belief over the group policy* $\tilde{\pi}^{\mathcal{G}}_a(u^{\mathcal{G}}_{\text{env}}|\tilde{\tau}^{\mathcal{G}}_t)$. In order to minimize the probability of disagreement during decentralized group action selection, agents in $\mathcal{G}$ can perform *optimal correlated sampling* based on a shared random seed (Holenstein, 2007; Bavarian et al., 2016). For a formal definition of *probabilistic common knowledge*, please refer to Krasucki et al. (1991, Definitions 8 and 13).

**Learning under common knowledge (LuCK)** is a novel cooperative multi-agent reinforcement learning setting, where a Dec-POMDP is augmented by a common knowledge function $\mathcal{I}^{\mathcal{G}}$ (or *probabilistic* common knowledge function $\tilde{\mathcal{I}}^{\mathcal{G}}_a$). Groups of agents $\mathcal{G}$ can coordinate by learning policies that condition on their common knowledge. In this paper $\mathcal{I}^{\mathcal{G}}$ (or $\tilde{\mathcal{I}}^{\mathcal{G}}_a$) is fixed apriori, but it could also be learnt during training. The setting accommodates a wide range of real-world and simulated multi-agent tasks. Whenever a task is cooperative and learning is centralised, then agents can naturally learn suitable $\mathcal{I}^{\mathcal{G}}$ or $\tilde{\mathcal{I}}^{\mathcal{G}}_a$. Policy parameters can be exchanged during training as well and thus become part of the commonly known data $\xi$. Joint policies where coordinated decisions of a group $\mathcal{G}$ only condition on the common knowledge of $\mathcal{G}$ can be executed in a fully decentralised fashion. In Section 3 we introduce MACKRL, which uses centralised training to learn fully decentralised policies under common knowledge.

**Field-of-view common knowledge** is a form of *complete-history common knowledge* (Halpern & Moses, 2000), that arises within a Dec-POMDP if agents can deduce parts of other agents' observations from their own. In this case, an agent group's common knowledge is the intersection of observations that all members can reconstruct from each other. In Appendix E we formalise this concept and show that, under some assumptions, common knowledge is the intersection of all agents' sets of visible objects, if and only if all agents can see each other. Figure 1 shows an example for three agents with circular fields of view. If observations are noisy, each agent bases its belief on its own noisy observations thus inducing an equivalent form of probabilistic common knowledge $\tilde{\mathcal{I}}^{\mathcal{G}}_a$.

Field-of-view common knowledge naturally occurs in many interesting real-world tasks, such as autonomous driving (Cao et al., 2013) and robo-soccer (Genter et al., 2017), as well as in simulated benchmarks such as StarCraft II (Vinyals et al., 2017). A large number of cooperative multi-agent tasks can therefore benefit from common knowledge-based coordination introduced in this paper.

## 3 Multi-Agent Common Knowledge Reinforcement Learning

The key idea behind MACKRL is to learn decentralised policies that are nonetheless coordinated by common knowledge. As the common knowledge history $\tau^{\mathcal{G}}_t$ of a group of agents $\mathcal{G}$ can be deduced by

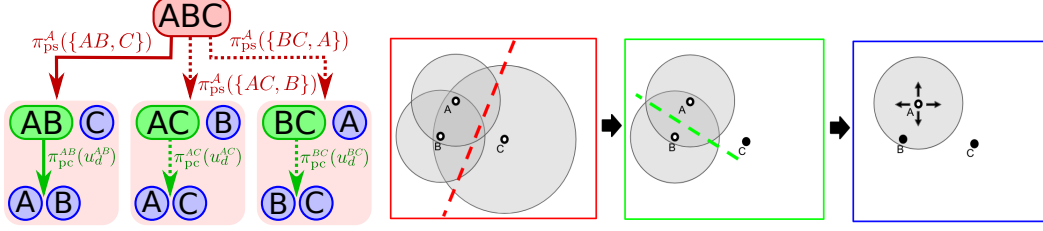

Figure 2: An illustration of Pairwise MACKRL. [left]: the full hierarchy for 3 agents (dependencies on common knowledge are omitted for clarity). Only solid arrows are computed during decentralised sampling with Algorithm 1, while all arrows must be computed recursively during centralised training (see Algorithm 2). [right]: the (maximally) 3 steps of decentralised sampling from the perspective of agent A.

1. Pair selector $\pi_{\text{ps}}^{\mathcal{A}}$ chooses the partition $\{AB, C\}$ based on the common knowledge of all agents $\mathcal{I}^{ABC}(\tau^A, \xi) = \varnothing$.

2. Based on the common knowledge of pair $A$ and $B$, $\mathcal{I}^{AB}(\tau^A, \xi)$, the pair controller $\pi_{\text{pc}}^{AB}$ can either choose a joint action $(u_{\text{env}}^A, u_{\text{env}}^B)$, or delegate to individual controllers by selecting $u_d^{AB}$.

3. If delegating, the individual controller $\pi^A$ must select the action $u_{\text{env}}^A$ for the single agent $A$. All steps can be computed based on A's history $\tau^A$.

every member, i.e., $\tau_t^{\mathcal{G}} = \mathcal{I}^{\mathcal{G}}(\tau_t^a, \xi), \forall a \in \mathcal{G}$, any deterministic function based only on $\tau_t^{\mathcal{G}}$ can thus be independently computed by every member as well. The same holds for pseudo-random functions like stochastic policies, if they condition on a commonly known random seed in $\xi$.

MACKRL uses a hierarchical policy $\pi(\mathbf{u}_{\text{env}} | \{\tau_t^a\}_{a \in \mathcal{A}}, \xi)$ over the joint environmental action space of all agents $\mathcal{U}_{\text{env}}$. The hierarchy forms a tree of sub-policies $\pi^{\mathcal{G}}$ over groups $\mathcal{G}$, where the root $\pi^{\mathcal{A}}$ covers all agents. Each sub-policy $\pi^{\mathcal{G}}(u^{\mathcal{G}} | \mathcal{I}^{\mathcal{G}}(\tau_t^{\mathcal{G}}, \xi))$ conditions on the common knowledge of $\mathcal{G}$, including a shared random seed in $\xi$, and can thus be executed by every member of $\mathcal{G}$ independently. The corresponding action space $\mathcal{U}^{\mathcal{G}}$ contains the environmental actions of the group $\mathbf{u}_{\text{env}}^{\mathcal{G}} \in \mathcal{U}_{\text{env}}^{\mathcal{G}}$ *and/or* a set of group partitions, that is, $\mathbf{u}^{\mathcal{G}} = \{\mathcal{G}_1, \dots, \mathcal{G}_k\}$ with $\mathcal{G}_i \cap \mathcal{G}_j = \varnothing$, $\forall i \neq j$ and $\cup_{i=1}^k \mathcal{G}_i = \mathcal{G}$.

Choosing a partition $\mathbf{u}^{\mathcal{G}} \neq \mathcal{U}_{\text{env}}^{\mathcal{G}}$ yields control to the sub-policies $\pi^{\mathcal{G}_i}$ of the partition's subgroups $\mathcal{G}_i \in \mathbf{u}^{\mathcal{G}}$. This can be an advantage in states where the common history $\tau_t^{\mathcal{G}_i}$ of the subgroups is more informative than $\tau_t^{\mathcal{G}}$. All action spaces have to be specified in advance, which induces the hierarchical tree structure of the joint policy. Algorithm 1 shows the decentralised sampling of environmental actions from the hierarchical joint policy as seen by an individual agent $a \in \mathcal{A}$.

As the common knowledge of a group with only one agent $\mathcal{G} = \{a\}$ is $\mathcal{I}^{\{a\}}(\tau^a, \xi) = \tau^a$, fully decentralised policies are a special case of MACKRL policies: in this case, the root policy $\pi^{\mathcal{A}}$ has only one action $\mathcal{U}^{\mathcal{A}} := \{\mathbf{u}^{\mathcal{A}}\}$, $\mathbf{u}^{\mathcal{A}} := \{\{1\}, \dots, \{n\}\}$, and all leaf policies $\pi^{\{a\}}$ have only environmental actions $\mathcal{U}^{\{a\}} := \mathcal{U}_{\text{env}}^a$.

## 3.1 Pairwise MACKRL

To give an example of one possible MACKRL architecture, we define *Pairwise MACKRL*, illustrated in Figure 2. As joint action spaces grow exponentially in the number of agents, we restrict ourselves to *pairwise* joint policies and define a three-level hierarchy of controllers.

---

**Algorithm 1** Decentralised action selection for agent $a \in \mathcal{A}$ in MACKRL

---

**function** SELECT_ACTION($a, \tau_t^a, \xi$)       $\triangleright$ random seed in $\xi$ is common knowledge
  $\mathcal{G} := \mathcal{A}$                       $\triangleright$ initialise the group $\mathcal{G}$ of all agents
  $\mathbf{u}_t^{\mathcal{G}} \sim \pi^{\mathcal{G}}\big(\cdot \,|\, \mathcal{I}^{\mathcal{G}}(\tau_t^a, \xi)\big)$       $\triangleright \mathbf{u}_t^{\mathcal{G}}$ is either a joint environmental action in $\mathcal{U}_{\text{env}}^{\mathcal{G}}$...
  **while** $\mathbf{u}_t^{\mathcal{G}} \notin \mathcal{U}_{\text{env}}^{\mathcal{G}}$ **do**       $\triangleright$ ... or a set of disjoint subgroups $\{\mathcal{G}_1, \dots, \mathcal{G}_k\}$
    $\mathcal{G} := \{\mathcal{G}' \,|\, a \in \mathcal{G}', \mathcal{G}' \in \mathbf{u}_t^{\mathcal{G}}\}$       $\triangleright$ select subgroup containing agent $a$
    $\mathbf{u}_t^{\mathcal{G}} \sim \pi^{\mathcal{G}}\big(\cdot \,|\, \mathcal{I}^{\mathcal{G}}(\tau_t^a, \xi)\big)$       $\triangleright$ draw an action for that subgroup
  **return** $u_t^a$       $\triangleright$ return environmental action $u_t^a \in \mathcal{U}_{\text{env}}^a$ of agent $a$

---

**Algorithm 2** Compute joint policies for a given $\mathbf{u}_{\text{env}}^{\mathcal{G}} \in \mathcal{U}_{\text{env}}^{\mathcal{G}}$ of a group of agents $\mathcal{G}$ in MACKRL

---

**function** JOINT_POLICY($\mathbf{u}_{\text{env}}^{\mathcal{G}} | \mathcal{G}, \{\tau_t^a\}_{a \in \mathcal{G}}, \xi$)  $\quad\triangleright$ random seed in $\xi$ is common knowledge

$\quad a' \sim \mathcal{G}$; $\quad \mathbf{I}^{\mathcal{G}} := \mathcal{I}^{\mathcal{G}}(\tau_t^{a'}, \xi) \quad\triangleright$ common knowledge $\mathbf{I}^{\mathcal{G}}$ is identical for every agent $a' \in \mathcal{G}$

$\quad p_{\text{env}} := 0 \quad\quad\quad\quad\quad\quad\quad\quad\quad\quad\triangleright$ initialise probability for choosing environmental joint action $\mathbf{u}_{\text{env}}^{\mathcal{G}}$

$\quad$**for** $\mathbf{u}^{\mathcal{G}} \in \mathcal{U}^{\mathcal{G}}$ **do** $\quad\quad\quad\quad\quad\quad\triangleright$ add probability to choose $\mathbf{u}_{\text{env}}^{\mathcal{G}}$ for all outcomes of $\pi^{\mathcal{G}}$

$\quad\quad$**if** $\mathbf{u}^{\mathcal{G}} = \mathbf{u}_{\text{env}}^{\mathcal{G}}$ **then** $\quad\quad\quad\quad\quad\triangleright$ if $\mathbf{u}^{\mathcal{G}}$ is the environmental joint action in question

$\quad\quad\quad p_{\text{env}} := p_{\text{env}} + \pi^{\mathcal{G}}(\mathbf{u}_{\text{env}}^{\mathcal{G}} | \mathbf{I}^{\mathcal{G}})$

$\quad\quad$**if** $\mathbf{u}^{\mathcal{G}} \notin \mathcal{U}_{\text{env}}^{\mathcal{G}}$ **then** $\quad\quad\quad\quad\quad\triangleright$ if $\mathbf{u}^{\mathcal{G}} = \{\mathcal{G}^1, \ldots, \mathcal{G}^k\}$ is a set of disjoint subgroups

$\quad\quad\quad p_{\text{env}} := p_{\text{env}} + \pi^{\mathcal{G}}(\mathbf{u}^{\mathcal{G}} | \mathbf{I}^{\mathcal{G}}) \prod\limits_{\mathcal{G}' \in \mathbf{u}^{\mathcal{G}}} \text{JOINT\_POLICY}(\mathbf{u}_{\text{env}}^{\mathcal{G}'} | \mathcal{G}', \{\tau_t^a\}_{a \in \mathcal{G}'}, \xi)$

$\quad$**return** $p_{\text{env}} \quad\quad\quad\quad\quad\quad\quad\quad\triangleright$ return probability that controller $\pi^{\mathcal{G}}$ would have chosen $\mathbf{u}_{\text{env}}^{\mathcal{G}}$

---

The root of this hierarchy is the *pair selector* $\pi_{\text{ps}}^{\mathcal{A}}$, with an action set $\mathcal{U}_{\text{ps}}^{\mathcal{A}}$ that contains all possible partitions of agents into pairs $\{\{a_1, \bar{a}_1\}, \ldots, \{a_{n/2}, \bar{a}_{n/2}\}\} =: \mathbf{u}^{\mathcal{A}} \in \mathcal{U}_{\text{ps}}^{\mathcal{A}}$, but no environmental actions. If there are an odd number of agents, then one agent is put in a singleton group. At the second level, each *pair controller* $\pi_{\text{pc}}^{a\bar{a}}$ of the pair $\mathcal{G} = \{a, \bar{a}\}$ can choose between joint actions $\mathbf{u}_{\text{env}}^{a\bar{a}} \in \mathcal{U}_{\text{env}}^a \times \mathcal{U}_{\text{env}}^{\bar{a}}$ and one delegation action $u_d^{a\bar{a}} := \{\{a\}, \{\bar{a}\}\}$, i.e., $\mathcal{U}_{\text{pc}}^{a\bar{a}} := \mathcal{U}_{\text{env}}^a \times \mathcal{U}_{\text{env}}^{\bar{a}} \cup \{u_d^{a\bar{a}}\}$. At the third level, individual controllers $\pi^a$ select an individual action $u_{\text{env}}^a \in \mathcal{U}_{\text{env}}^a$ for a single agent $a$. This architecture retains manageable joint action spaces, while considering all possible pairwise coordination configurations. Fully decentralised policies are the special case when all pair controllers always choose partition $u_d^{a\bar{a}}$ to delegate.

Unfortunately, the number of possible pairwise partitions is $O(n!)$, which limits the algorithm to medium sized sets of agents. For example, $n = 11$ agents induce $|\mathcal{U}_{\text{ps}}^{\mathcal{A}}| = 10395$ unique partitions. To scale our approach to tasks with many agents, we share network parameters between all pair controllers with identical action spaces, thereby greatly improving sample efficiency. We also investigate a more scalable variant in which the action space of the pair selector $\pi_{\text{ps}}^{\mathcal{A}}$ is only a fixed random subset of all possible pairwise partitions. This restricts agent coordination to a smaller set of predefined pairs, but only modestly affects MACKRL's performance (see Section 4.2).

### 3.2 Training

The training of policies in the MACKRL family is based on *Central-V* (Foerster et al., 2018), a stochastic policy gradient algorithm (Williams, 1992) with a centralised critic. Unlike the decentralised policy, we condition the centralised critic on the state $s_t \in \mathcal{S}$ and the last actions of all agents $\mathbf{u}_{\text{env},t-1} \in \mathcal{U}_{\text{env}}$. We do not use the multi-agent counterfactual baseline proposed by Foerster et al. (2018), because MACKRL effectively turns training into a single agent problem by inducing a correlated probability across the joint action space. Algorithm 2 shows how the probability of choosing a joint environmental action $\mathbf{u}_{\text{env}}^{\mathcal{G}} \in \mathcal{U}_{\text{env}}^{\mathcal{G}}$ of group $\mathcal{G}$ is computed: the probability of choosing the action in question is added to the recursive probabilities that each partition $\mathbf{u}^{\mathcal{G}} \notin \mathcal{U}_{\text{env}}^{\mathcal{G}}$ would have selected it. During decentralized execution, Algorithm 1 only traverses one branch of the tree. To further improve performance, all policies choose actions greedily outside of training and do thus not require any additional means of coordination such as shared random seeds during execution.

At time $t$, the gradient with respect to the parameters $\theta$ of the joint policy $\pi(\mathbf{u}_{\text{env}} | \{\tau_t^a\}_{a \in \mathcal{A}}, \xi)$ is:

$$\nabla_\theta J_t = \underbrace{\left( r(s_t, \mathbf{u}_{\text{env},t}) + \gamma V(s_{t+1}, \mathbf{u}_{\text{env},t}) - V(s_t, \mathbf{u}_{\text{env},t-1}) \right)}_{\text{sample estimate of the advantage function}} \nabla_\theta \log \big( \underbrace{\pi(\mathbf{u}_{\text{env},t} | \{\tau_t^a\}_{a \in \mathcal{A}}, \xi)}_{\text{JOINT\_POLICY}(\mathbf{u}_{\text{env},t} | \mathcal{A}, \{\tau_t^a\}_{a \in \mathcal{A}}, \xi)} \big), \quad (1)$$

The value function $V$ is learned by gradient descent on the TD($\lambda$) loss (Sutton & Barto, 1998). As the hierarchical MACKRL policy tree computed by Algorithm 2 is fully differentiable and MACKRL trains a joint policy in a centralised fashion, the standard convergence results for actor-critic algorithms (Konda & Tsitsiklis, 1999) with compatible critics (Sutton et al., 1999) apply.

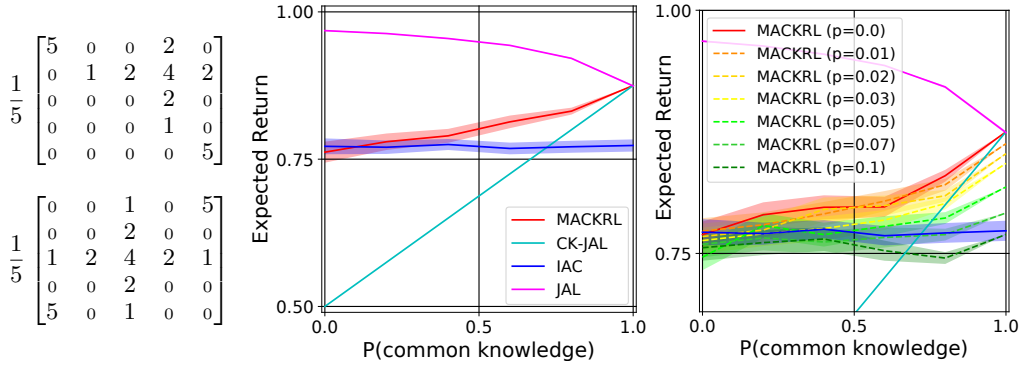

Figure 3: Game matrices A (top) and B (bottom) [left]. MACKRL almost always outperforms both IL and CK-JAL and is upper-bounded by JAL [middle]. When the common knowledge is noised by randomly flipping the CK-bit with probability $p$, MACKRL degrades gracefully [right].

## 4 Experiments and Results

We evaluate Pairwise MACKRL (henceforth referred to as MACKRL) on two environments[3]: first, we use a matrix game with special coordination requirements to illustrate MACKRL's ability to surpass both IL and JAL. Secondly, we employ MACKRL with deep recurrent neural network policies in order to outperform state-of-the-art baselines on a number of challenging StarCraft II unit micromanagement tasks. Finally, we demonstrate MACKRL's robustness to sensor noise and its scalability to large numbers of agents to illustrate its applicability to real-world tasks.

### 4.1 Single-step matrix game

To demonstrate how MACKRL trades off between independent and joint action selection, we evaluate a two-agent matrix game with partial observability. In each round, a fair coin toss decides which of the two matrix games in Figure 3 [left] is played. Both agents can observe which game has been selected if an observable *common knowledge bit* is set. If the bit is not set, each agent observes the correct game only with probability $p_\sigma$, and is given *no observation* otherwise. Crucially, whether agents can observe the current game is in this case determined independently of each other. Even if both agents can observe which game is played, this observation is no longer common knowledge and cannot be used to infer the choices of the other agent. To compare various methods we adjusted $p_\sigma$ such that the independent probability of each agent to observe the current game is fixed at 75%.

In order to illustrate MACKRL's performance, we compare it to three other methods: Independent Actor Critic (IAC, Foerster et al., 2018) is a variant of Independent Learning where each agent conditions both its decentralized actor and critic only on its own observation. Joint Action Learning (JAL, Claus & Boutilier, 1998) learns a centralized joint policy that conditions on the union of both agent's observations. CK-JAL is a decentralised variant of JAL in which both agents follow a joint policy that conditions only on the common knowledge available.

Figure 3 [middle] plots MACKRL's performance relative to IL, CK-JAL, and JAL against the fraction of observed games that are caused by a set CK-bit. As expected, the performance of CK-JAL linearly increases as more common knowledge becomes available, whereas the performance of IAC remains invariant. MACKRL's performance matches the one of IAC if no common knowledge is available and matches those of JAL and CK-JAL in the limit of all observed games containing common knowledge. In the regime between these extremes, MACKRL outperforms both IAC and CK-JAL, but is itself upper-bounded by JAL, which gains the advantage due to central execution.

To assess MACKRL's performance in the case of *probabilistic common knowledge* (see Section 2), we also consider the case where the observed *common knowledge bit* of individual agents is randomly flipped with probability $p$. This implies that both agents do not share true common knowledge with respect to the game matrix played. Instead, each agent $a$ can only form a belief $\tilde{\mathcal{I}}_a^{\mathcal{G}}$ over what is

commonly known. The commonly known pair controller policy can then be conditioned on each agent's belief, resulting in agent-specific pair controller policies $\tilde{\pi}_{pc,a}^{aa'}$.

As $\tilde{\pi}_{pc,a}^{aa'}$ and $\tilde{\pi}_{pc,a'}^{aa'}$ are no longer guaranteed to be consistent, agents need to sample from their respective pair controller policies in a way that minimizes the probability that their outcomes disagree in order to maximise their ability to coordinate. Using their access to a shared source of randomness $\xi$, the agents can optimally solve this *correlated sampling* problem using Holenstein's strategy (see Appendix D). However, this strategy requires the evaluation of a significantly larger set of actions and quickly becomes computationally expensive. Instead, we use a suboptimal heuristic that nevertheless performs satisfactorily in practice and can be trivially extended to groups of more than two agents: given a shared uniformly drawn random variable $\delta \sim \xi, 0 \leq \delta < 1$, each agent $a$ samples an action $u_a$ such that

$$\sum_{u=1}^{u_a-1} \tilde{\pi}_{pc,a}^{aa'}(u) \leq \delta < \sum_{u=1}^{u_a} \tilde{\pi}_{pc,a}^{aa'}(u). \tag{2}$$

Figure 3 [right] shows that MACKRL's performance declines remarkably gracefully with increasing observation noise. Note that as real-world sensor observations tend to tightly correlate with the true observations, noise levels of $p \geq 0.1$ in the context of the single-step matrix game are rather extreme in comparison, as they indicate a completely different game matrix. This illustrates MACKRL's applicability to real-world tasks with noisy observation sensors.

## 4.2 StarCraft II micromanagement

To demonstrate MACKRL's ability to solve complex coordination tasks, we evaluate it on a challenging multi-agent version of StarCraft II (SCII) micromanagement. To this end, we report performance on three challenging coordination tasks from the established multi-agent benchmark SMAC (Samvelyan et al., 2019).

The first task, map *2s3z*, contains mixed unit types, where both the MACKRL agent and the game engine each control two Stalkers and three Zealots. Stalkers are ranged-attack units that take heavy damage from melee-type Zealots. Consequently, a winning strategy needs to be able to dynamically coordinate between letting one's own Zealots attack enemy Stalkers, and when to backtrack in order to defend one's own Stalkers against enemy Zealots. The challenge of this coordination task results in a particularly poor performance of Independent Learning (Samvelyan et al., 2019).

The second task, map *3m*, presents both sides with three Marines, which are medium-ranged infantry units. The coordination challenge on this map is to reduce enemy fire power as quickly as possible by focusing unit fire to defeat each enemy unit in turn. The third task, map *8m*, scales this task up to eight Marines on both sides. The relatively large number of agents involved poses additional scalability challenges.

On all maps, the units are subject to partial observability constraints and have a circular field of view with fixed radius. Common knowledge $\mathcal{I}^{\mathcal{G}}$ between groups $\mathcal{G}$ of agents arises through entity-based field-of-view common knowledge (see Section 2 and Appendix E).

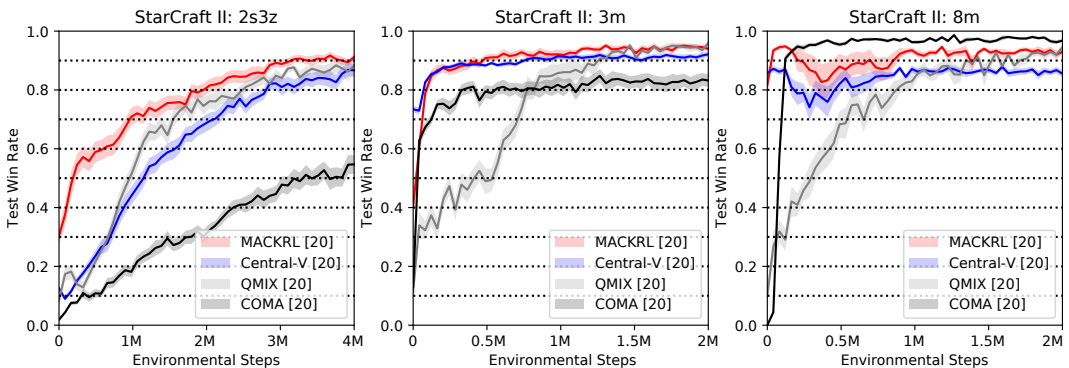

Figure 4: Win rate at test time across StarCraft II scenarios: 2 Stalkers & 3 Zealots [left], 3 Marines [middle] and 8 Marines [right]. Plots show means and their standard errors with [number of runs].

We compare MACKRL to Central-V (Foerster et al., 2018), as well as COMA (Foerster et al., 2018) and QMIX (Rashid et al., 2018), where the latter is an off-policy value-based algorithm that is the current state-of-the-art on all maps. We omit IL results since it is known to do comparatively poorly (Samvelyan et al., 2019). All experiments use SMAC settings for comparability (see Samvelyan et al. (2019) and Appendix B for details). In addition, MACKRL and its within-class baseline Central-V share equal hyper-parameters as far as applicable.

MACKRL outperforms the Central-V baseline in terms of sample efficiency and limit performance on all maps (see Figure 4). All other parameters being equal, this suggests that MACKRL's superiority over Central-V is due to its ability to exploit common knowledge. In Appendix C, we confirm this conclusion by showing that the policies learnt by the pair controllers are almost always preferred over individual controllers whenever agents have access to substantial amounts of common knowledge.

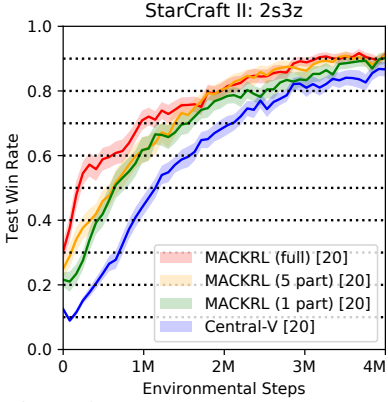

MACKRL also significantly outperforms COMA and QMIX on all maps in terms of sample efficiency, with a similar limit performance to QMIX (see Figure 4). These results are particularly noteworthy as MACKRL employs neither a sophisticated multi-agent baseline, like COMA, nor an off-policy replay buffer, like QMIX.

Figure 5: Illustrating MACKRL's scalability properties using partition subsamples of different sizes.

As mentioned in Section 3.1, the number of possible agent partitions available to the pair selector $\pi^{\mathcal{A}}_{\text{ps}}$ grows as $\mathcal{O}(n!)$. We evaluate a scalable variant of MACKRL that constrains the number of partitions to a fixed subset, which is drawn randomly before training. Figure 5 shows that sample efficiency declines gracefully with subsample size. MACKRL's policies appear able to exploit any common knowledge configurations available, even if the set of allowed partitions is not exhaustive.

## 5 Related Work

*Multi-agent reinforcement learning* (MARL) has been studied extensively in small environments (Busoniu et al., 2008; Yang & Gu, 2004), but scaling it to large state spaces or many agents has proved problematic. Guestrin et al. (2002a) propose the use of *coordination graphs*, which exploit conditional independence properties between agents that are captured in an undirected graphical model, in order to efficiently select joint actions. *Sparse cooperative Q-learning* (Kok & Vlassis, 2004) also uses coordination graphs to efficiently maximise over joint actions in the $Q$-learning update rule. Whilst these approaches allow agents to coordinate optimally, they require the coordination graph to be known and for the agents to either observe the global state or to be able to freely communicate. In addition, in the worst case there is no conditional independence to exploit and maximisation must still be performed over an intractably large joint action space.

There has been much work on scaling MARL to handle complex, high dimensional state and action spaces. In the setting of fully centralised training and execution, Usunier et al. (2016) frame the problem as a *greedy MDP* and train a centralised controller to select actions for each agent in a sequential fashion. Sukhbaatar et al. (2016) and Peng et al. (2017) train factorised but centralised controllers that use special network architectures to share information between agents. These approaches assume unlimited bandwidth for communication.

One way to decentralise the agents' policies is to learn a separate $Q$-function for each agent as in *Independent Q-Learning* (Tan, 1993). Foerster et al. (2017) and Omidshafiei et al. (2017) examine the problem of instability that arises from the nonstationarity of the environment induced by both the agents' exploration and their changing policies. Rashid et al. (2018) and Sunehag et al. (2017) propose learning a centralised value function that factors into per-agent components. Gupta et al. (2017) learn separate policies for each agent in an actor-critic framework, where the critic for each agent conditions only on per-agent information. Foerster et al. (2018) and Lowe et al. (2017) propose a single centralised critic with decentralised actors. None of these approaches explicitly learns a policy over joint actions and hence are limited in the coordination they can achieve.

Thomas et al. (2014) explore the psychology of common knowledge and coordination. Rubinstein (1989) shows that any finite number of reasoning steps, short of the infinite number required for common knowledge, can be insufficient for achieving coordination (see Appendix E). Korkmaz et al. (2014) examine common knowledge in scenarios where agents use Facebook-like communication. Brafman & Tennenholtz (2003) use a common knowledge protocol to improve coordination in common interest stochastic games but, in contrast to our approach, establish common knowledge about agents' action sets and not about subsets of their observation spaces.

Aumann et al. (1974) introduce the concept of a *correlated equilibrium*, whereby a shared *correlation device* helps agents coordinate better. Cigler & Faltings (2013) examine how the agents can reach such an equilibrium when given access to a simple shared *correlation vector* and a communication channel. Boutilier (1999) augments the state space with a coordination mechanism, to ensure coordination between agents is possible in a fully observable multi-agent setting. This is in general not possible in the partially observable setting we consider.

Amato et al. (2014) propose MacDec-POMDPs, which use hierarchically optimal policies that allow agents to undertake temporally extended macro-actions. Liu et al. (2017) investigate how to learn such models in environments where the transition dynamics are not known. Makar et al. (2001) extend the MAXQ single-agent hierarchical framework (Dieterich, 2000) to the multi-agent domain. They allow certain policies in the hierarchy to be *cooperative*, which entails learning the joint action-value function and allows for faster coordination across agents. Kumar et al. (2017) use a hierarchical controller that produces subtasks for each agent and chooses which pairs of agents should communicate in order to select their actions. Oh & Smith (2008) employ a hierarchical learning algorithm for cooperative control tasks where the outer layer decides whether an agent should coordinate or act independently, and the inner layer then chooses the agent's action accordingly. In contrast with our approach, these methods require communication during execution and some of them do not test on sequential tasks.

Nayyar et al. (2013) show that common knowledge can be used to reformulate decentralised planning problems as POMDPs to be solved by a central coordinator using dynamic programming. However, they do not propose a method for scaling this to high dimensions. By contrast, MACKRL is entirely model-free and learns trivially decentralisable control policies end-to-end.

Guestrin et al. (2002b) represent agents' value functions as a sum of context-specific value rules that are part of the agents' fixed a priori common knowledge. By contrast, MACKRL learns such value rules dynamically during training and does not require explicit communication during execution.

Despite using a hierarchical policy structure, MACKRL is not directly related to the family of hierarchical reinforcement learning algorithms (Vezhnevets et al., 2017), as it does not involve temporal abstraction.

## 6   Conclusion and Future Work

This paper proposed a way to use common knowledge to improve the ability of decentralised policies to coordinate. To this end, we introduced MACKRL, an algorithm which allows a team of agents to learn a fully decentralised policy that nonetheless can select actions jointly by using the common knowledge available. MACKRL uses a hierarchy of controllers that can either select joint actions for a pair or delegate to independent controllers.

In evaluation on a matrix game and a challenging multi-agent version of StarCraft II micromanagement, MACKRL outperforms strong baselines and even exceeds the state of the art by exploiting common knowledge. We present approximate versions of MACKRL that can scale to greater numbers of agents and demonstrate robustness to observation noise.

In future work, we would like to further increase MACKRL's scalability and robustness to sensor noise, explore off-policy variants of MACKRL and investigate how to exploit limited bandwidth communication in the presence of common knowledge. We are also interested in utilising SIM2Real transfer methods (Tobin et al., 2017; Tremblay et al., 2018) in order to apply MACKRL to autonomous car and unmanned aerial vehicle coordination problems in the real world.

**Acknowledgements**

We would like to thank Chia-Man Hung, Tim Rudner, Jelena Luketina, and Tabish Rashid for valuable discussions. This project has received funding from the European Research Council (ERC) under the European Union's Horizon 2020 research and innovation programme (grant agreement number 637713), the National Institutes of Health (grant agreement number R01GM114311), EPSRC/MURI grant EP/N019474/1 and the JP Morgan Chase Faculty Research Award. This work is linked to and partly funded by the project Free the Drones (FreeD) under the Innovation Fund Denmark and Microsoft. It was also supported by the Oxford-Google DeepMind Graduate Scholarship and a generous equipment grant from NVIDIA.

## Footnotes

[3]All source code is available at https://github.com/schroederdewitt/mackrl.

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
