[Supplementary Material · MACKRL_appendix.pdf]

# Appendix

## A  Pairwise MACKRL

This section aims to give a better intuition about Pairwise MACKRL using the example of three agents $\mathcal{A} := \{1, 2, 3\}$. The explicitly written-out joint policy of Pairwise MACKRL for these agents is:

$$
\begin{aligned}
\boldsymbol{\pi}_\theta(u_{\mathrm{env}}^1, u_{\mathrm{env}}^2, u_{\mathrm{env}}^3) = {} & \pi_{\mathrm{ps},\theta}^{\mathcal{A}}(u_{\mathrm{ps}}^{\mathcal{A}} {=} \{\{1\}, \{2,3\}\} | \mathcal{I}_s^{1,2,3}) \cdot \pi_\theta^1(u_{\mathrm{env}}^1 | \tau^1) \\
& \cdot \Big( \pi_{\mathrm{pc},\theta}^{2,3}(u_{\mathrm{env}}^{2,3} | \mathcal{I}_s^{2,3}) + \pi_{\mathrm{pc},\theta}^{2,3}(u_d^{2,3} | \mathcal{I}_s^{2,3}) \cdot \pi_\theta^2(u_{\mathrm{env}}^2 | \tau^2) \cdot \pi_\theta^3(u_{\mathrm{env}}^3 | \tau^3) \Big) \\[6pt]
+ {} & \pi_{\mathrm{ps},\theta}^{\mathcal{A}}(u_{\mathrm{ps}}^{\mathcal{A}} {=} \{\{2\}, \{1,3\}\} | \mathcal{I}_s^{1,2,3}) \cdot \pi_\theta^2(u_{\mathrm{env}}^2 | \tau^2) \\
& \cdot \Big( \pi_{\mathrm{pc},\theta}^{1,3}(u_{\mathrm{env}}^{1,3} | \mathcal{I}_s^{1,3}) + \pi_{\mathrm{pc},\theta}^{1,3}(u_d^{1,3} | \mathcal{I}_s^{1,3}) \cdot \pi_\theta^1(u_{\mathrm{env}}^1 | \tau^1) \cdot \pi_\theta^2(u_{\mathrm{env}}^3 | \tau^3) \Big) \\[6pt]
+ {} & \pi_{\mathrm{ps},\theta}^{\mathcal{A}}(u_{\mathrm{ps}}^{\mathcal{A}} {=} \{\{3\}, \{1,2\}\} | \mathcal{I}_s^{1,2,3}) \cdot \pi_\theta^3(u_{\mathrm{env}}^3 | \tau^3) \\
& \cdot \Big( \pi_{\mathrm{pc},\theta}^{1,2}(u_{\mathrm{env}}^{1,2} | \mathcal{I}_s^{1,2}) + \pi_{\mathrm{pc},\theta}^{1,2}(u_d^{1,2} | \mathcal{I}_s^{1,2}) \cdot \pi_\theta^1(u_{\mathrm{env}}^1 | \tau^1) \cdot \pi_\theta^2(u_{\mathrm{env}}^2 | \tau^2) \Big) .
\end{aligned}
$$

Conditional variables beyond common knowledge $\mathcal{I}^{\mathcal{G}}$ and action-observation histories $\tau^a$ have been omitted for brevity. See Table 1 for a detailed depiction of Pairwise MACKRL's hierarchical controllers.

| Level | Policy / Controller | #$\pi$ |
|-------|---------------------|--------|
| 1 | $\pi_{\mathrm{ps}}(u^{\mathrm{ps}} | \mathcal{I}_{s_t}^{\mathcal{A}}, u_{t-1}^{\mathrm{ps}}, h_{t-1}^{\mathrm{ps}})$ | 1 |
| 2 | $\pi_{\mathrm{pc}}^{aa'}(u^{aa'} | \mathcal{I}_{s_t}^{aa'}, u_{t-1}^{aa'}, h_{t-1}^{aa'}, aa')$ | 3 |
| 3 | $\pi^a(u^a | z_t^a, h_{t-1}^a, u_{t-1}^a, a)$ | 3 |

Table 1: Hierarchy of pairwise MACKRL, where $h$ is the hidden state of RNNs and $z_t^a$ are observations. #$\pi$ shows the number of controllers at this level for 3 agents.

However the sampling of each agent's actions $u_{\mathrm{env}}^a \in \mathcal{U}^a$ only needs to traverse one branch of the tree, as shown in Figure 6. At the top level, an agent id partition $u_{\mathrm{ps}}$ is categorically sampled from the pair selector policy $\pi_{\mathrm{ps},\theta}$. At the second level, the pair selector policy for the pair contained in the partition $\pi_{\mathrm{pc}}^{a,b}$ is categorically sampled from in order to receive $u^{a,b}$ where $a, b \in u_{\mathrm{ps}}$. If the delegation action $d$ is sampled, then both $u^a$ and $u^b$ are categorically resampled from their respective independent policies $\pi^a$ and $\pi^b$. Otherwise, $u^a$ and $u^b$ are determined by $u^{a,b}$. The leftover agent $c \notin u_{\mathrm{ps}}$ samples its action from its corresponding independent policy $\pi^c$. Note that this sampling scheme naturally generalised for $n > 3$.

Figure 6: Action sampling for MACKRL for $n = 3$ agents.

$c_1, c_2 = \mathcal{CK}; \sigma_1, \sigma_2 = A$

$\begin{cases} A, A & \text{with prob. } p_\sigma^2 \\ A, ? & \text{with prob. } (1 - p_\sigma)\, p_\sigma \\ ?, A & \text{with prob. } p_\sigma\, (1 - p_\sigma) \\ ?, ? & \text{with prob. } (1 - p_\sigma)^2 \end{cases}$

$c_1, c_2 = \mathcal{CK}; \sigma_1, \sigma_2 = B$

$\begin{cases} B, B & \text{with prob. } p_\sigma^2 \\ B, ? & \text{with prob. } (1 - p_\sigma)\, p_\sigma \\ ?, B & \text{with prob. } p_\sigma\, (1 - p_\sigma) \\ ?, ? & \text{with prob. } (1 - p_\sigma)^2 \end{cases}$

Figure 7: Probability tree for our simple single-step matrix game. The game chooses randomly between matrix $A$ or $B$, and whether common knowledge is available or not. If common knowledge is available, both agents can condition their actions on the game matrix chosen. Otherwise, both agents independently only have a random chance of observing the game matrix choice. Here, $p_{ck}$ is the probability that common knowledge exists and $p_\sigma$ is the probability that an agent independently observes the game matrix choice. The observations of each agent 1 and 2 are given by tuples $(c_1, \sigma_1)$ and $(c_2, \sigma_2)$, respectively, where $c_1, c_2 \in \{\mathcal{CK}, \mathcal{\cancel{CK}}\}$ and $\sigma_1, \sigma_2 \in \{A, B, ?\}$.

## B  Experimental Setup - StarCraft II

All policies are implemented as two-layer recurrent neural networks (GRUs) with 64 hidden units, while the critic is feed forward and uses full state information. Parameters are shared across controllers within each of the second and third levels of the hierarchy. We also feed into the policy the agent index or index pairs. For exploration, we use a bounded softmax distribution in which the agent samples from a softmax over the policy logits with probability $(1 - \epsilon)$ and samples randomly with probability $\epsilon$. We anneal $\epsilon$ from $0.5$ to $0.01$ across the first 50k environment steps.

Episodes are collected using eight parallel SCII environments. Optimisation is carried out on a single GPU with Adam and a learning rate of $0.0005$ for both the agents and the critic. The policies are fully unrolled and updated in a large mini-batch of $T \times B$ entries, where $T = 60$ and $B = 8$. By contrast, the critic is optimised in small mini-batches of size 8, one for each time-step, looping backwards in time. We found that this stabilised and accelerated training compared to full batch updates for the critic. The target network for the critic is updated after every 200 critic updates. We use $\lambda = 0.8$ in TD($\lambda$) to accelerate reward propagation.

## C  Pair controller introspection

We now test the hypothesis that MACKRL's superior performance is indeed due to its ability to learn how to use common knowledge for coordination. To demonstrate that the pair controller can indeed learn to delegate strategically, we plot in Figure 8 the percentage of delegation actions $u_d$ against the number of enemies in the common knowledge of the selected pair controller, in situations where there is at least some common knowledge.

Figure 8: Delegation rate vs. number of enemies (2s3z) in the common knowledge of the pair controller over training.

Since we start with randomly initialised policies, at the beginning of training the pair controller delegates only rarely to the decentralised controllers. As training proceeds, it learns to delegate in most situations where the number of enemies in the common knowledge of the pair is small, the exception being no visible enemies, which happens too rarely (5% of cases). This shows that MACKRL can learn to delegate in order

to take advantage of the private observations of the agents, but also learns to coordinate in the joint action space when there is substantial common knowledge.

## D  Holenstein's Strategy

Given two agent-specific pair controller policies $\tilde{\pi}_{pc,a}^{aa'}$, both agents can optimally minimise disagreement when sampling independently form their respective policies by following Holenstein's strategy (Holenstein, 2007; Bavarian et al., 2016): With a suitably chosen $\gamma > 0$, each agent $a$ is assigned a set

$$\mathcal{H}_a = \{(u,p) \in \mathcal{U}_{pc}^{aa'} \times \Gamma : p < \tilde{\pi}_{pc,a}^{aa'}(u)\}, \ \Gamma = \{0, \gamma, 2\gamma, \ldots, 1\} \tag{3}$$

Let $\zeta$ be a shared $\xi$-seeded random permutation of the elements in $\mathcal{U}_{pc}^{aa'} \times \Gamma$, then agent $a$ samples $\zeta(i_a)$, where $i_a$ is the smallest index such that $\zeta(i_a) \in \mathcal{H}_a$ (and agent $a'$ proceeds analogously). Given the total variational distance $\delta$ between the categorical probability distributions defined by $\tilde{\pi}_{pc,a}^{aa'}$ and $\tilde{\pi}_{pc,a'}^{aa'}$, the disagreement probability of agents $a, a'$ is then guaranteed to be at most $2\delta/(1 + \delta)$ (Bavarian et al., 2016).

## E  Common Knowledge with Entities

To exploit a particular form of *field-of-view common knowledge* with MACKRL, we formalise an instance of a Dec-POMDP, in which such common knowledge naturally arises. In this Dec-POMDP, the state $s$ is composed of a number of entities $e \in \mathcal{E}$, with state features $s^e$, i.e., $s = \{s^e \,|\, e \in \mathcal{E}\}$. Some entities are agents $a \in \mathcal{A} \subseteq \mathcal{E}$. Other entities could be enemies, obstacles, or goals.

The agents have a particular form of partial observability: the observation $z^a$ contains the subset of state features $s^e$ from all the entities $e$ that $a$ can see. Whether $a$ can observe $e$ is determined by the binary mask $\mu^a(s^a, s^e) \in \{\top, \bot\}$ over the agent's and entity's observable features. An agent can always observe itself, i.e., $\mu^a(s^a, s^a) = \top, \forall a \in \mathcal{A}$. The set of all entities the agent can see is therefore $\mathcal{M}^a := \{e \,|\, \mu^a(s^a, s^e)\} \subseteq \mathcal{E}$, and the agent's observation is specified by the deterministic observation function $o(s, a)$ such that $z^a = o(s, a) = \{s^e \,|\, e \in \mathcal{M}^a\} \in \mathcal{Z}$. In the example of Figure 1, $\mathcal{M}^A = \mathcal{M}^B = \{A, B\}$ and $\mathcal{M}^C = \{A, B, C\}$.

This special Dec-POMDP yields perceptual aliasing in which the state features of each entity are either accurately observed or completely occluded. The Dec-POMDP could be augmented with additional state features that do not correspond to entities, as well as additional possibly noisy observation features, without disrupting the common knowledge we establish about entities. For simplicity, we omit such additions.

A key property of the binary mask $\mu^a$ is that it depends only on the features $s^a$ and $s^e$ to determine whether agent $a$ can see entity $e$. If we assume that an agent $a$'s mask $\mu^a$ is common knowledge, then this means that another agent $b$, that can see $a$ and $e$, i.e., $a, e \in \mathcal{M}^b$, can deduce whether $a$ can also see $e$. This assumption can give rise to common knowledge about entities. Figure 1 demonstrates this for 3 agents with commonly known observation radii.

The *mutual knowledge* $\mathcal{M}^{\mathcal{G}}$ of a group of agents $\mathcal{G} \subseteq \mathcal{A}$ in state $s$ is the set of entities that all agents in the group can see in that state: $\mathcal{M}^{\mathcal{G}} := \cap_{a \in \mathcal{G}} \mathcal{M}^a$. However, mutual knowledge does not imply common knowledge. Instead, the *common knowledge* $\mathcal{I}^{\mathcal{G}}$ of group $\mathcal{G}$ in state $s \in \mathcal{S}$ is the set of entities such that all agents in $\mathcal{G}$ see $\mathcal{I}^{\mathcal{G}}$, know that all other agents in $\mathcal{G}$ see $\mathcal{I}^{\mathcal{G}}$, know that they know that all other agents see $\mathcal{I}^{\mathcal{G}}$, and so forth (Osborne & Rubinstein, 1994).

To know that another agent $b$ also sees $e \in \mathcal{E}$, agent $a$ must see $b$ and $b$ must see $e$, i.e., $\mu^a(s^a, s^b) \wedge \mu^b(s^b, s^e)$. Common knowledge $\mathcal{I}^{\mathcal{G}}$ can then be formalised recursively for every agent $a \in \mathcal{G}$ as:

$$\mathcal{I}_0^a := \mathcal{M}^a, \qquad \mathcal{I}_m^a := \bigcap_{b \in \mathcal{G}} \{e \in \mathcal{I}_{m-1}^b \,|\, \mu^a(s^a, s^b)\}, \qquad \mathcal{I}^{\mathcal{G}} := \lim_{m \to \infty} \mathcal{I}_m^a. \tag{4}$$

This definition formalises the above description that common knowledge is the set of entities that a group member sees ($m = 0$), that it knows all other group members see as well ($m = 1$), and so forth ad infinitum. In the example of Figure 1, $\mathcal{I}^{AB} = \{A, B\}$ and $\mathcal{I}^{AC} = \mathcal{I}^{BC} = \mathcal{I}^{ABC} = \varnothing$.

The following lemma establishes that, in our setting, if a group of agents can all see each other, their common knowledge is their mutual knowledge.

**Lemma 1.** *In the setting described in this Section, and when all masks are known to all agents, the common knowledge of a group of agents $\mathcal{G}$ in state $s \in \mathcal{S}$ is*

$$\mathcal{I}^{\mathcal{G}} = \begin{cases} \mathcal{M}^{\mathcal{G}}, & \text{if } \bigwedge_{a,b \in \mathcal{G}} \mu^a(s^a, s^b) \\ \varnothing, & \text{otherwise} \end{cases} . \tag{5}$$

*Proof.* The lemma follows by induction on $m$. The recursive definition of common knowledge (4) holds trivially if $\mathcal{I}^{\mathcal{G}} = \varnothing$. Starting from the knowledge of any agent $a$ in state $s$, $\mathcal{I}_0^a = \mathcal{M}^a$, definition (4) yields:

$$\mathcal{I}_1^a = \begin{cases} \mathcal{M}^{\mathcal{G}}, & \text{if } \bigwedge_{b \in \mathcal{G}} \mu^a(s^a, s^b) \\ \varnothing, & \text{otherwise} \end{cases} .$$

Next we show inductively that if all agents in group $\mathcal{G}$ know the mutual knowledge $\mathcal{M}^{\mathcal{G}}$ of state $s$ at some iteration $m$, that is, $\mathcal{I}_m^c \stackrel{\text{ind.}}{=} \mathcal{M}^{\mathcal{G}}$, then this mutual knowledge becomes common knowledge two iterations later. Applying the definition (4) for any agent $a \in \mathcal{G}$ twice yields:

$$\mathcal{I}_{m+2}^a = \bigcap_{b \in \mathcal{G}} \bigcap_{c \in \mathcal{G}} \left\{ e \in \mathcal{I}_m^c \Big| \mu^a(s^a, s^b) \wedge \mu^b(s^b, s^c) \right\} = \left\{ e \in \mathcal{E} \Big| \bigwedge_{b \in \mathcal{G}} \left( \mu^a(s^a, s^b) \wedge \bigwedge_{c \in \mathcal{G}} \left( \mu^b(s^b, s^c) \wedge e \in \mathcal{I}_m^c \right) \right) \right\}$$

$$\stackrel{\text{ind.}}{=} \left\{ e \in \mathcal{M}^{\mathcal{G}} \Big| \bigwedge_{b,c \in \mathcal{G}} \mu^b(s^b, s^c) \right\},$$

which is the right side of (5), and where we used

$$\bigwedge_{b \in \mathcal{G}} \left( \mu^a(s^a, s^b) \wedge \bigwedge_{c \in \mathcal{G}} \mu^b(s^b, s^c) \right) = \bigwedge_{b,c \in \mathcal{G}} \mu^b(s^b, s^c), \ \forall a \in \mathcal{G}.$$

Finally, applying (4) one more time to this result, yields:

$$\mathcal{I}_{m+3}^a = \bigcap_{b \in \mathcal{G}} \left\{ e \in \mathcal{I}_{m+2}^b \Big| \mu^a(s^a, s^b) \right\} = \mathcal{I}_{m+2}^a .$$

For all $m \geq 3$, $\mathcal{I}_m^a$ remains thus the right hand side of (5). As $\mathcal{I}^{\mathcal{G}} = \lim_{m \to \infty} \mathcal{I}_m^a$, we can thus conclude that, starting at the knowledge of any agent of group $\mathcal{G}$, in which all agents can see each other, the mutual knowledge is the common knowledge. □

The common knowledge can be computed using only the visible set $\mathcal{M}^a$ of every agent $a \in \mathcal{G}$. Moreover, actions that have been chosen by a policy, which itself is common knowledge, and that further depends only on common knowledge and a shared random seed, are also common knowledge. The common knowledge of group $\mathcal{G}$ up to time $t$ is thus some common prior knowledge $\xi$ and the commonly known trajectory $\tau_t^{\mathcal{G}} = (\xi, z_1^{\mathcal{G}}, \mathbf{u}_1^{\mathcal{G}}, \ldots, z_t^{\mathcal{G}}, \mathbf{u}_t^{\mathcal{G}})$, with $z_k^{\mathcal{G}} = \{ s_k^e \,|\, e \in \mathcal{I}^{\mathcal{G}} \}$. Knowing all binary masks $\mu^a$ makes it possible to derive $\tau_t^{\mathcal{G}} = \mathcal{I}^{\mathcal{G}}(\tau^a, \xi)$ from the observation trajectory $\tau_t^a = (z_1^a, \ldots, z_t^a)$ of any agent $a \in \mathcal{G}$ and the shared prior knowledge $\xi$. A function that conditions on $\tau^{\mathcal{G}}$ can therefore be computed independently by every member of $\mathcal{G}$.

Note that (by definition) common knowledge can only arise from entities that are observed *identically* by all agents. If only one agent receives non-deterministic observations, for example induced by sensor noise, the other agents cannot deduce the group's mutual (and thus common) knowledge. Our method therefore only guarantees perfect decentralisation of the learned policy in settings with deterministic observations, like simulations and computer games. However, in Section 4.1 we show empirically that, using a naive correlated sampling protocol similar to the theoretically optimal Holenstein protocol (Holenstein, 2007; Bavarian et al., 2016), MACKRL can still succeed in the presence of moderate sensor noise.