[Reviews · NeurIPS 2019]

Reviewer 1



My two biggest complaints center on 1) the illustrative single-step matrix game of section 4.1 and figure 3 and 2) the practical applications of MACKRL. 1) Since the primary role of the single-step matrix game in section 4.1 is illustrative, it should be much clearer what is going on. How are all 3 policies parameterized? What information does each have access to? What is the training data? First, let's focus on the JAL policy. As presented up until this point in the paper, JAL means centralized training *and* execution. If one is in a setting where centralized execution is practical, then JAL can be used and should perform at least as well as MACKRL. The advantage of MACKRL is that it can be applied where dencentralized execution is required. Therefore, the advantage of MACKRL is in wider applicability, not in performance on equal footing. So what is happening in Figure 3, middle? As far as I can reverse engineer from the results, the "JAL" model here can only leverage common knowledge, that is it only knows the matrix game when the common knowledge bit is set to 1. It does not receive access to the second, independent coin flip that the other two models have access to. I find this quite misleading. Intuitively, in a JAL model, any knowledge that *one* agent has should be available to *all* agents; that is, all knowledge is common, since execution is centralized. That is clearly not what was done here. This is important, because as presented, the implication is that MACKRL is better than JAL in a direct comparison, whereas I believe the advantage of MACKRL is in being able to be applied in settings where JAL cannot. Second, let's focus on the IAC policy. It is confusing that performance increases as p(common knowledge) increases, since independent agents cannot leverage this. Of course, the reason is because, as p(common knowledge) increases, so does p(independent knowledge). This is a confusing representation. I believe the results would be better presented if parameterized such that these two numbers varied independently. That is, one could imagine flipping coins for the two agents independently, and then with probability p letting them look at each other's coins. Then, the probability that an agent knows the game is decorrelated from whether there is common knowledge, and I think the results would be more clearly presented this way. In any case, as parameterized, the IAC results are still strange. For example, why do the IAC agents not achieve 100% performance when p(common knowledge)=1? In this case, the game is always revealed to both agents. Of course, they do not "know" the other agent knows the game, but even treating the other agent as part of the environment, learning an optimal strategy here should be easy. For example, in game 1, the agents could blindly coordinate on choosing action 1 (or 5), and in game 2, one would "specialize" in action 1 and the other in action 5. This can be learned without modeling the other agent. So why didn't the IAC agents learn this? Well, they probably would have if they were *only* trained with p(common knowledge)=1. I have to assume the authors trained the same agents over the whole range of p(common knowledge), so the task has a sort of hidden meta learning element of it that isn't quite stated explicitly. This should be. 2) I agree with the authors that common knowledge is an interesting and important concept for coordinating multi-agent learning. However, as presented, it seems somewhat brittle. First, for any (pseudo-)stochastic elements of the (hierarchical or primitive action) policies, the agents must have access to a shared random seed. But how is this coordinated? If all agents are deployed at the same time and either perform the same number of draws from the random number generator or have access to a synchronized clock, depending on how the RNG is implemented, then this should work fine. However, one imagines that, in reality, often agents will be deployed at different times and perhaps execute actions and therefore random number draws and not quite exactly synchronized intervals. Thus, the assumption of access to shared random number draws could quickly break down. That said, if this is the biggest challenge, perhaps that isn't so bad, because communicating a single scalar among a group of (mostly) dencentralized agents doesn't sound too challenging in many domains. Still, I wish this had been addressed in the paper. Second, the bar for common knowledge in this setting seems quite high, since it is assumed to be infinitely recursive and certain, that is knowledge that I 100% know that you 100% know that I 100% know and so on. Of course, many (most?) forms of common knowledge fall short of this. The knowledge may be finite-order (e.g. I know that you know, but not beyond that) or held with uncertainty. The authors try to address the latter with Figure 3 right, but its not clear to me what I should learn from this. Performance seems to degrade to the level of independent learners after just 10% corruption. Also I don't know what their "correlated sampling" approach is and the authors don't explain in the main text (another sentence or two in the main text would be nice). I would like to have seen independent agents *infer* what other agents know (both from direct observations and reasoning about behavior) and then to act upon that (probabilistic) common knowledge. Some conceptual clarifications: 3) Figure 1 - agent C can see both A and B and see that they see each other. Can agent C use the knowledge that A and B may/will make a joint decision in its own decision? As I understand the framework presented, the answer is "no", but could it be expanded to allow for this? 4) line 142 - I can imagine situations where, conditioning only on common knowledge, a pair of agents might choose a joint decision. However, one agent might have private knowledge that makes it clear they will do better acting independently. For example, perhaps two agents are working together to lift a rock to retrieve food (the rock is too heavy for just one agent alone to lift, so they must coordinate). They are looking at each other and the intersection of their fields of vision is the small space between them. Conditioning on this shared field of view, continuing the present activity is the best decision. However, a man-eating tiger is creeping up behind one of the agents, in full view of the other agent. It is clear the two agents should instead run or defend themselves in some way, but since the tiger's presence is not yet common knowledge, the greedy operation of the MACKRL hierarchical policy will probably choose the joint action. Am I understanding this correctly? Should MACKRL include an "independent override" ability for agents to reject the joint action and make an independent one when the expected value difference is very high? 5) line 86 - the setting here is fully cooperative with shared reward and centralized training. Could a variant of MACKRL be useful when agents incentives are not fully aligned? More minor comments and questions: 6) paragraph 1 - capitalization of "Joint action learning" and "Independent Learning" should be matched 7) sentence on lines 36-38 - would be nice to have an example or citation 8) line 36 - "... MACKRL uniquely occupies a middle ground between IL and JAL..." - This seems way too strong. Plenty of methods fall in this middle ground - anything that models other agents explicitly instead of treating then as part of the environment (e.g. Raileanu et al 2018 Modeling Others using Oneself in Multi-Agent Reinforcement Learning, which should probably also be cited). 9) lines 80-81: why are action spaces agent-specific but not observation spaces? (U_env has an "a" superscript, but not Z) 10) line 94 - \citet -> \citep 11) line 97 and 101 - please be more clear on what kind of mathematical object tau_t^G is. It is for example unclear what \geq means here. 12) line 163 - other heuristic one might leverage is greedy pairwise selection based on spatial proximity 13) line 173 and eqn 1 - why is the state value function conditioned on last actions as well? This doesn't seem standard. 14) eqn 1 - J_t should be introduced in words. 15) Figure 4 - number of runs shouldn't be in the legend since it is the same in all cases. Remove it from legend and put it once in the caption. 16) Figure 5 - in either the caption or main text discussion, the authors might want to comment on the total number of partitions the "full" version is choosing from, to place the restricted selector version in context. 17) line 295 - missing a space between sentences 18) citations - Rabinowitz et al 2018 Machine Theory of Mind focuses on inferring what agents know from observing their behavior. Relevant to inferred common knowledge. UPDATE: Thanks to the authors for their helpful rebuttal. In particular, thanks for clarifying the matrix game example; I think the presentation is much clearer now. I've raised my score from a 6 to a 7.

Reviewer 2



The rebuttal addresses my concern on the matrix game. I encourage the authors to improve the presentation of the matrix game example in the final version. ---- The paper proposes a framework for multi-agent RL that relies on common knowledge. Overall I consider it to be a high-quality paper. The idea of applying common knowledge to multi-agent is important and interesting, and this paper takes a solid step in this direction with modern deep learning techniques. The framework is well presented with a specific architecture that considers using common knowledge in pairwise agents. The paper is overall well written, but I do have some questions/confusions listed below. Regarding the results in Figure 3 (middle). It is surprising to me that JAL does not dominate IAC. It would be much better if the authors explicitly illustrate the policies that those methods learn when P(common knowledge) is 0, 0.5, and 1.0, respectively. Also, visually the expected return of MACKRL is linear when P(common knowledge)>0.5 but wagging when <0.5. JAL is linear and IAC is wagging in [0,1]. This phenomenon is non-trivial and deserves some explanation from the authors. The last paragraph of Section 3.1 is confusing. It reads “we first of all share network parameters between *all pair selectors* with identical action spaces to greatly increase sample efficiency”. Isn’t it true that we have *only one* pair selector here which is the root of the policy tree. What parameters are we sharing?

Reviewer 3



This paper provides a novel model named multi-agent common knowledge reinforcement learning (MACKRL) which aims to learn a hierarchical policy tree. Also, the authors provide pairwise MACKRL as an example of one possible MACKRL architecture. The paper is well-organized and clearly presented. However, there are several concerns that the authors need to further improve: 1. Model Efficiency: As the authors claim, the possible pairwise partitions is O(n!), all possible partitions including tuple-wise cases may reach O(n^n). As the number of agents increases, the model suffers greatly from exponentially-growing model complexity although the authors restrict themselves to pairwise setting. 2. Ablation Study: As the author state in Algorithm 1, the common knowledge is a random seed and participates in choosing partition, deciding action types and taking action. However, the authors do not verify whether the common knowledge is actually working or ignored as the similar cases in Computer Vision fields. The experiments for ablation study are expected to add. 3. Experiments with HRL: As the authors state, MACKRL uses a hierarchical policy over the joint environment action space. Considering several hierarchical reinforcement learning (HRL) methods also deal with the similar task, the additional experiments for further comparison with these HRL methods like FuN are expected.

[Author Response · NeurIPS 2019]

We would like to thank all reviewers for their time and helpful comments.

**Revised matrix game (@R1,2)** We are sorry that the matrix game caused confusion and will use this feedback to
make the exposition more intuitive. Briefly: all policies use tabular representations, for reward matrices see Fig 3, for
policy inputs and game details see Fig 7 in Supplement. We will add details on the training procedure.

*JAL*: Indeed, a confusing name. Since we focus on decentralized execution, we used a JAL that only conditions on CK,
which we will rename 'CK-JAL'. We have added a true JAL which receives the joint observation. As expected this
upper-bounds performance.

*IAC*: Independent learners converge to a local optimum (e.g. rows 2 and 3 for agent A) when the other agent is not
reliably coordinating for the global maximum (as will be the case during training). So even when the state is known to
both agents, the coordination problem leads to suboptimal IAC performance. All $P(CK)$ are trained separately and
there is no meta/transfer learning. MACKRL gets around the IAC limitation by allowing agents to explore and learn in
the joint action space using CK; it does not suffice for either agent to know independently which game is played. In
contrast to IAC, MACKRL and JAL will specialize on actions 1 and 5 (as suggested by R1).

*Presentation of results*: R1 makes a great suggestion, we have now changed the game such that the unconditional
probability (i.e. $p(\texttt{independent knowledge})$) of an agent observing the game is fixed at 75%. On the x-axis, we
change which % of these observations is due to CK and which is due to independent observations. Note that the JAL
benefits from having a low % of CK observations, since it's less likely that neither of the agents observes the state. @R2
- we are investigating the wagging which likely stems from sub-optimal tuning of the gradient-based optimiser.

**Practical applications and scaling (@R1,3)** We use deterministic policies in
decentralised execution so there is no need for any further coordination. If we
wish to use a stochastic policy, it is easy to share a random seed either from the
centralised training phase, or using, e.g., synchronised clocks or a highly limited
communication channel. Deploying agents at different times is a different setting
(ad hoc teamwork) that we don't address in this work.

While the formal definition of CK is strict, in practice, humans easily relax these
requirements to perform commonsense reasoning. If Alice tells Bob, "Meet me
at King's Cross" there's no point for Alice to actually go there without *assuming*
the CK that Bob heard and understood what Alice said, and the CK concerning
social conventions. We find that MACKRL, perhaps surprisingly, is quite robust
to a naive relaxation to a type of probabilistic CK. Note that in a normal sensor,
noisy observations will be closely correlated with the true observation. In our
game, by contrast, 10% bit flip noise is quite extreme, indicating a completely different matrix. We will elaborate in the
paper on softer forms of CK (see e.g. Halpern & Moses 2000) and our use of correlated sampling. We are also excited
about future work which would extend MACKRL even more robustly to these settings.

Any MARL approach faces challenges in scaling with the number of agents. What matters is how they address it:
MACKRL does so with a sampling approach that we show is quite effective (Fig 5), more so than independent learning
(which scales well, but at the cost of highly limited capacity for coordination).

**Baselines and ablations (@R3)** HRL methods (including FuN) target temporal abstraction, and are not relevant or
comparable to MACKRL. We will clarify that 'policy hierarchy' in no way refers to HRL. In Appendix C, we perform
one possible introspective study, showing the pair controllers prefer not to delegate when coordination is difficult and
CK is large (i.e. when many enemies are present in the CK).

**Further clarifications** We will update the paper to reflect the following and address all other minor issues raised.

(R1-3) Nothing disallows C from using its knowledge about A and B's CK, if the pair controller delegates to C. However,
it is not commonly known by A and B that C knows about their CK.

(R1-4) An "independent override" heuristic is an interesting avenue for future research which we believe lies outside the
scope of this paper. We predict the current form of MACKRL would simply learn to delegate to the individual agents in
order to mitigate the risk of the tiger.

(R1-5) If agents incentives are not fully aligned, then establishing conventions is more difficult because agents may
have incentives to violate the conventions. Nonetheless, coordination devices play an important role in game theory and
CK could help agents to coordinate in these settings. However, this is well beyond the scope of this paper.

(R1-8) Should read "a unique middle ground" rather than "uniquely..."

(R1-9) In our setting the action space can depend on the unit-type of the agent and hence the agent ID, while the
observation space is common across all agents.

(R1-11) $\tau_t^{\mathcal{G}}$ is a set and line 101 should use $\supseteq$ rather than $\geq$. We will define $\tau_t^{\mathcal{G}}$ more precisely.

(R1-12) Pairwise selection based on proximity is an interesting avenue, but note that the selection itself needs to be
based on CK (which doesn't in general include global proximity information).

(R2) S3.1 should read that 'pair controllers' have shared parameters.

[Meta-Review · NeurIPS 2019]

All reviewers agreed this paper is well written presenting some interesting novel ideas. Reviewers believe that integrating common knowledge directly into Multi-agent RL training is a nice idea, and suggests some interesting future directions of research. Initially, there were shared concerns and confusion though about a number of issues, most prominently about the matrix game example. After reading and discussing the authors rebuttal though it seems the authors adequately addressed some of the primary concerns, and the general sense is that this paper is solid and of interest to be presented at NeurIPS.